# Evaluation of Terrestrial Water Storage Changes and Major Driving Factors Analysis in Inner Mongolia, China

**DOI:** 10.3390/s22249665

**Published:** 2022-12-09

**Authors:** Yi Guo, Fuping Gan, Baikun Yan, Juan Bai, Naichen Xing, Yue Zhuo

**Affiliations:** 1China Aero Geophysical Survey and Remote Sensing Center for Natural Resources, China Geological Survey, Beijing 100083, China; 2Key Laboratory of Aerial Geophysics and Remote Sensing Geology, Ministry of Natural Resources, Beijing 100083, China

**Keywords:** GRACE, land surface model, TWS, GWS, climate changes, agricultural irrigation, coal mining, Inner Mongolia

## Abstract

Quantitative assessment of the terrestrial water storage (TWS) changes and the major driving factors have been hindered by the lack of direct observations in Inner Mongolia, China. In this study, the spatial and temporal changes of TWS and groundwater storage (GWS) in Inner Mongolia during 2003–2021 were evaluated using the satellite gravity data from the Gravity Recovery and Climate Experiment (GRACE) and the GRACE Follow On combined with data from land surface models. The results indicated that Inner Mongolia has experienced a widespread TWS loss of approximately 1.82 mm/yr from 2003–2021, with a more severe depletion rate of 4.15 mm/yr for GWS. Meteorological factors were the driving factors for water storage changes in northeastern and western regions. The abundant precipitation increased TWS in northeast regions at 2.36 mm/yr. Anthropogenic activities (agricultural irrigation and coal mining) were the driving factors for water resource decline in the middle and eastern regions (especially in the agropastoral transitional zone), where the decrease rates were 4.09 mm/yr and 3.69 mm/yr, respectively. In addition, the severities of hydrological drought events were identified based on water storage deficits, with average severity values of 17 mm, 18 mm, 24 mm, and 33 mm for the west, middle, east, and northeast regions, respectively. This study established a basic framework for water resource changes in Inner Mongolia and provided a scientific foundation for further water resources investigation.

## 1. Introduction

Climate changes have triggered observable perturbations to the regional water cycle and water storage, which have been further intensified by human activities [1,2,3]. Water storage depletion has become a regional or global threat to the sustainable development of human society and ecosystems, especially in arid and semi-arid regions [4,5,6,7]. Terrestrial water storage (TWS) is the total water in the land, including surface water, soil water, groundwater, etc. [8]. TWS plays an important role in the water cycle, energy cycle, and biochemical cycle, and is very sensitive to environmental changes. Therefore, it is of great significance to estimate the spatial and temporal changes of the TWS, quantify the extent of hydrological droughts, and evaluate the major driving factors for better water resource management in arid and semi-arid regions.

The Gravity Recovery and Climate Experiment (GRACE), launched on 17 March 2002, measured the Earth’s gravity field changes [9,10,11]. The time-variable gravity field data of GRACE have been used to reveal the effects of earthquakes down to a magnitude of 8.3 on the earth’s gravity [12,13,14] to estimate ice mass losses [15,16,17], evaluate the sea-level changes [18,19,20], and quantify the TWS changes [8,21,22,23]. Although GRACE failed to work since October 2017, the successor GRACE Follow On (GRACE-FO) launched in 2018 has been an alternative for the following time-variable gravity field data [24,25]. (GRACE terminology will be used for both GRACE and GRACE-FO missions in the article).

TWS changes based on GRACE have two major advantages in hydrological studies. Firstly, the GRACE-based TWS changes are integrated changes in all forms of water [8]. Therefore, GRACE data could be used to quantify groundwater storage (GWS) changes [26,27], runoff discharge [28,29,30], and evapotranspiration [31,32,33] with the assistance of land surface models (LSMs). Secondly, GRACE-based TWS changes capture the comprehensive responses to both natural environmental changes and anthropogenic activities [34]. For some regions, climate changes have been regarded as the dominant factor controlling water storage [35,36], while in other regions, the direct and indirect consequences of anthropogenic activities can also be linked to water shortage decreases [37,38]. In addition, GRACE data extend a new way to reflect the water deficit and provide additional information about the effects of meteorological drought, and quantify the severity of hydrological drought [39,40,41,42,43].

Despite the above advantages, it cannot be denied there are limitations to studying water resource changes by GRACE and LSMs. The coarse spatial resolution of GRACE data is the primary factor affecting the TWS changes at the basin scale [44,45]. LSMs also have certain limitations in simulating the soil moisture in deep soil layers [46]. However, the study of Swenson et al. [47] showed that if the study area reached more than 40 × 10^4^ km^2^, the error of TWS changes based on GRACE data would be less than 1 cm, and the larger the study area, the higher the accuracy. In addition, recent studies have demonstrated that water storage changes inferred from the GRACE data and LSMs have sufficient resolution for regional water resource management [25,48,49,50].

Inner Mongolia Autonomous Region (Inner Mongolia hereafter) covers 12% of China’s land area, while its water resources only account for 1.76% of the total water resources in China. The deficit of water resources has become an important factor in the restoration of the ecological environment in Inner Mongolia. Previous studies have preliminarily explored the water resource status and supply/demand balance [51,52,53,54]. However, there is a lack of studies on the regional water resource changes in Inner Mongolia and responses to climate changes and human activities. This is mainly because Inner Mongolia spans several climate zones, which results in complex climate-hydrological conditions. Therefore, the objectives of this study were (1) to quantify the spatiotemporal variations of TWS in different climate zones in Inner Mongolia over 2003–2021; (2) to identify the impacts of climate changes and anthropogenic activities on TWS changes in Inner Mongolia; and (3) to evaluate the hydrological droughts based on water storage deficit in Inner Mongolia. This study could provide a basic framework for water resource changes in Inner Mongolia and the responses of its water resource changes to climate changes and human activities, which are beneficial to water resource management and ecological restoration.

## 2. Materials and Methods

### 2.1. Study Area

The Inner Mongolia Autonomous Region, referred to as “Inner Mongolia”, straddles the northeast, north, and northwest regions of China (37°24′–53°23′ N, 97°12′–126°04 E). The area of Inner Mongolia is 118.3 × 10^4^ km^2^, accounting for 12.3% of the total land area of China [51]. Inner Mongolia is connected to 8 provinces (autonomous regions), borders Mongolia and Russia, and plays an important role in ensuring national defense security and regional stability [52]. The topography has a slope from northeast to southwest, with a long and narrow shape (Figure 1a). Inner Mongolia has a unique ecosystem structure of forests, grasslands, agropastoral ecotone, and deserts due to the long east-west span (Figure 1b).

Inner Mongolia has an arid/semi-arid continental monsoon climate, with dry and windy springs, hot and rainy summers, cool autumns, and a prolonged cold period in winter, with characteristics of distinct seasons and synchronous rain and heat [53]. Considering the vast territory of Inner Mongolia, climate conditions in different regions are rather different. According to the China climatic regionalization data, Inner Mongolia is divided into four climatic zones: The west climate zone, middle climate zone, east climate zone, and northeast climate zone (Geographic Data Sharing Infrastructure, College of Urban and Environmental Science, Peking University (http://geodata.pku.eud.cn/, accessed on 13 October 2022).

Inner Mongolia has well-developed strata, frequent magmatic activity, good metallogenic conditions, and rich mineral resources [55]. Inner Mongolia is one of the largest coal producers, ranks second in coal production, and has the largest number of open-pit coal mines [55]. There are Paleozoic Carboniferous Permian coalfields and Mesozoic Jurassic coalfields in the middle of Inner Mongolia. There are two tectonic units, the Tianshan-Inner Mongolia-Xing’an geosyncline in the north and the North China platform in the south. Influenced by the westward subduction of the Pacific plate during the Mesozoic and Cenozoic, a NEE trending tectonic volcanic belt was formed in the eastern part of Inner Mongolia.

The total water resources in Inner Mongolia are only 50.93 billion m^3^ (37.13 billion for surface water and 13.80 billion m^3^ for groundwater), only accounting for 1.86% of the total water resources in China. The main characteristics of water resources in Inner Mongolia are as follows: (1) Water resources are extremely uneven in space and restricted by natural conditions [53]; (2) water resources show regional shortages due to low precipitation and large evapotranspiration [54]; and (3) water resource utilization does not adapt to the strategic layout requirements of regional economic and social development [52].

### 2.2. Data

#### 2.2.1. GRACE Mascon Data

The GRACE mascon solutions provided by the Center for Space Research, University of Texas (CSR-M), were used in this study. The mascon solutions (CSR-M) applied the time variable regularization method to constrain the inversion of satellite-ranging rates to gravity fields without the requirement of post-processing and scale factors correction [56]. The TWS was computed as anomalies (TWSA) relative to the 2004–2009 mean baseline in equivalent water height. The temporal resolution was monthly, and the spatial resolution was 1° × 1°.

#### 2.2.2. GLDAS Data

Three LSMs (CLSM, NOAH, and VIC) in the Global Land Data Assimilation System (GLDAS) were used in this study [57]. The temporal resolution and spatial resolution of the three LSMs were the same as in the GRACE mascon solution. The soil moisture (SM), snow depth water equivalent (SWE), and plant canopy surface water (CanoInt) were used in this study to estimate the GWS changes.

#### 2.2.3. Other Auxiliary Data

The gridded precipitation data were obtained from the Global Precipitation Measurement (GPM) [58]. In order to improve the data accuracy, the in situ precipitation data from the China Meteorological Data Service Center were used to correct the GPM data. The evapotranspiration data were from the Global Land Evaporation Amsterdam Model (GLEAM) [59,60]. The human water consumption data, including agricultural consumption, were obtained from water resource bulletins or statistical yearbooks [61,62].

### 2.3. Methods

#### 2.3.1. Groundwater Changes Based on Water Mass Balance Approach

Under the assumption that the water storage changes of rivers, lakes, and reservoirs are negligible, GWS anomalies (GWSA) were calculated as the following equation:GWSA = TWSA − SMA − SWEA − CanoIntA(1)

TWSA was from GRACE, and SMA, SWEA, and CanoIntA were the anomalies of SM, SWE, and CanoInt from GLDAS. The SM was provided by NOAH, which was the summation of soil moisture in four soil layers (0–10 cm, 10–40 cm, 40–100 cm, and 100–200 cm). The SWE provided by three LSMs showed a high correlation, and SWE values in this study were the mean values of SWE from the above-mentioned three LSMs. The CanoInt provided by CLSM and VIC showed a high correlation, while CanoInt values from NOAH were significantly higher than those of the other two models. Therefore, CanoInt values were the mean values of CLSM and VIC. To unify the parameters, all variables were converted to mm in equivalent water height, and then the SMA, SWEA, and CanoIntA were calculated by subtracting each set of monthly data from the 2004–2009 mean value.

#### 2.3.2. Water Storage Deficit Index Based on GRACE

The total water storage deficit (WSD) was calculated by the methods of Thomas et al. [43] and Sun et al. [39] to identify the drought events.
(2)$$WSD=TWSA_{i,j}-\bar{TWSA_{i,j}}$$
where $TWSA_{i,j}$ is the TWSA for the *j*th month in the *i*th year and $\bar{TWSA_{i,j}}$ is the mean value of TWSA for the *j*th month.

According to Thomas et al. [43], the water storage deficit severity (Se) was used to calculate the water needed to return to normal water resource conditions, which was the product of WSD and the duration in months ($D\left(t\right)$).
(3)$$Se=WSD\times D\left(t\right)$$

#### 2.3.3. The Mann–Kendall Trend Test

The Mann–Kendall trend test method is a reliable method for obtaining the trend of hydro-meteorological time series. The Mann–Kendall trend test was performed on the grid annual TWSA, SMA, and GWSA to reveal the spatial change rates. The original time series ($X_{t}$) was divided by the mean of the sequence, and a new sequence ($Y_{t}$) with an average value of 1 was created. The trend estimator β of the new sequence was calculated [63,64]:(4)$$\beta \;=\;\mathrm{Median}\left(\frac{y_{i}-y_{j}}{i-j}\right)\;\;\left(1\leq i\leq j\leq n\right)$$
where $y_{i}$ and $y_{j}$ are the new sequence at time of i and j, β < 0 showed a decreasing trend, and β > 0 showed an increasing trend.

#### 2.3.4. The Cross-Correlation Analysis

The cross-correlation analysis can reveal the relationship between two sequences in multiple time lags. The cross-correlation coefficient (CCF) was applied to quantify the statistical relationship between different time series ($x_{t}$ and $y_{t}$) [65]:(5)$$r\left(k\right)=\frac{1}{n}\overset{n-k}{\underset{1}{\sum }}(x_{t}-\bar{x})\left(y_{t+k}-\bar{y}\right)$$
(6)$$CCF=\frac{r\left(k\right)}{\sigma_{x}\sigma_{y}}$$
where $x_{t}$ is the hydro-climate variables at the time of *t*, $y_{t+k}$ is the TWS changes at the lag time of *k* corresponding to $x_{t}$, $\bar{x}$ and $\bar{y}$ are the mean values of the time series of $x_{t}$ and $y_{t}$, and $\sigma_{x}$ ($\sqrt{\frac{1}{n}\sum_{1}^{n}{\left(x_{t}-\bar{x}\right)}^{2}}$) and $\sigma_{y}$ ($\sqrt{\frac{1}{n}\sum_{1}^{n}{\left(y_{t}-\bar{y}\right)}^{2}})$ are the standard deviations of the time series $x_{t}$ and $y_{t}$.

#### 2.3.5. The Time Series Decomposition

The time series was decomposed into the long-term trend and seasonal change signals as follows [66]:(7)$$y_{t}=a+bt+csin\left(\frac{2\pi t}{12}\right)+dcos\left(\frac{2\pi t}{12}\right)+esin\left(\frac{2\pi t}{6}\right)+fcos\left(\frac{2\pi t}{6}\right)$$
where a is the constant term, b is the long-term trend, c and d are annual terms, and e and f are semi-annual terms. The amplitudes and phases of annual and semi-annual terms were $A_{ann}$, $\theta_{ann}$, $A_{semi-ann}$, and $\theta_{semi-ann}$, respectively.
(8)$$A_{ann}=\sqrt{c^{2}+d^{2}},\;\theta_{ann}=tan^{-1}\frac{d}{c}$$
(9)$$A_{semi-ann}=\sqrt{e^{2}+f^{2}},\;\theta_{semi-ann}=tan^{-1}\frac{f}{e}$$

Desmoothing was applied for time series containing a long-term trend and periodic seasonal components as the time averaging would influence the amplitudes [67]. The amplitudes were restored by multiplying the estimated annual amplitude in (8) by 1.0115 and the semi-annual amplitude in (9) by 1.0472 in accordance with [67].

## 3. Results

### 3.1. The Temporal Changes of TWS and Its Hydrological Components

#### 3.1.1. The Water Storage Changes in Inner Mongolia

Figure 2 shows the time series of monthly TWSA and its hydrological components in Inner Mongolia from 2003–2021. In this period, the TWSA in Inner Mongolia decreased at the rate of 1.82 mm/yr (2.15 Gt/yr); the SMA increased at the rate of 2.15 mm/yr (2.54 Gt/yr); the SWEA exhibited no significant trend. Correspondingly, the GWSA decreased at the rate of 4.15 mm/yr (4.91 Gt/yr) based on the water balance (Figure 2).

#### 3.1.2. The Water Storage Changes in Different Climate Zones

In order to reveal the influences of climate conditions on water resource changes, TWSA and its hydrological components of four climate zones were compared (Figure 3). The details were as below:In the west climate zone, TWSA decreased at the rate of 3.12 mm/yr from 2003–2021 (Figure 3a). The SMA was stable with the least amplitude among the four climate zones. The SWEA was the lowest among the four climate regions. Based on the water balance, the GWSA decreased at the rate of 3.61 mm/yr from 2003–2021.In the middle climate zone, the TWSA decreased at a rate of 4.09 mm/yr from 2003 to 2021. The amplitude of SMA was the largest (272.14 mm) among the four climate zones. The SMA was stable from 2003–2016 and increased at the rate of 39.02 mm/yr in 2017–2021 (Figure 3b). The GWSA decreased at the rate of 9.25 mm/yr from 2003–2021 and 47.37 mm/yr from 2018–2021.In the east climate zone, the TWSA decreased at a rate of 3.69 mm/yr from 2003 to 2021. In fact, the TWSA changed in stages; specifically, it decreased at the rate of 7.71 mm/yr from January 2003 to April 2011 and 11.83 mm/yr from May 2012 to April 2018 and increased at the rate of 22.01 mm/yr from May 2018 to December 2021. The amplitude of SMA followed the middle climate zone with 216.26 mm. The SMA was relatively stable from 2003 to 2016 and increased at the rate of 12.01 mm/yr in 2017–2021. The GWSA decreased at the rate of 6.15 mm/yr in 2003–2021.Differently from the three above-mentioned climate zones, the TWSA increased at the rate of 2.36 mm/yr from 2003 to 2021 in the northeast climate zone. The TWSA showed obvious stages, with three increase stages (2008.2–2009.9/2012.2–2013.12/2018.2–2021.12) and three decrease stages (2004.5–2007.10/2010.5–2011.10/2013.12–2018.1). From 2003 to 2017, SMA was stable with seasonal fluctuations, and increased at the rate of 9.93 mm/yr from 2018 to 2021. The SWEA was the highest among the four climate regions (Figure 3c). The GWSA increased at the rate of 0.94 mm/yr in 2003–2021 with a significance level of 0.1 (the level of significance in this study was 0.05 without special description), at the rate of 35.05 mm/yr in 2018–2021.

### 3.2. The Spatial Changes of Tws and Its Hydrological Components

Figure 4 shows the spatial distribution of the annual average TWSA and its hydrological components from 2003–2021. There was a surplus of TWSA in northeastern regions and a deficit in most regions of Inner Mongolia; there was also a surplus of SMA in most of Inner Mongolia, and SWEA was negligible compared to TWSA and SMA. Correspondingly, GWSA displayed a deficit in most regions and a surplus only in a small part of the northeastern regions. The box diagrams show the changes in soil moisture and groundwater were obvious compared with the surface water changes (the sum of SWE changes and CanoInt changes) (Figure 5). Therefore, soil moisture and groundwater were the major contributors to TWS changes.

## 4. Discussion

### 4.1. The Responses of Terrestrial Water Resources to Climate Changes

From the perspective of intensive climate changes and human activities, it is of great importance to analyze the major driving factors of water resource changes. According to the NCEP-NCAR reanalysis data [68], the air temperature decreased at the rate of 0.18 °C/10a in 1948–1996 with an average of 5.95 °C and increased at the rate of 0.24 °C/10a in 1997–2020 with an average of 6.49 °C. Correspondingly, from the 1960s to the 1990s, precipitation fluctuated relatively little except in some special years; from the middle of the 1990s to the 2000s, the fluctuation of precipitation increased on the whole [53]. According to the GPM data, the average annual precipitation was 288 mm in Inner Mongolia and showed an increasing fluctuation during the period of 2003–2021. The southeast air with the Pacific water vapor was the main precipitation source of Inner Mongolia. As the water vapor moved from southeast to northeast, it was blocked by the Helan Mountains and Yin Mountains to the west and the Greater Khingan Mountains to the north, forming a small numbeer of precipitation resources with an uneven distribution. The annual precipitation was relatively abundant for the northeast climate zone and the east climate zone at 275–628 mm and 263–536 mm, respectively. The precipitation was 161–399 in the middle climate zone with an average value of 275 mm. In particular, the precipitation was less than 100 mm in most of the west climate zone, and the average annual precipitation was only 125 mm (Figure 6a). The average annual evapotranspiration also increased from 145 mm to 308 mm, 367 mm, and 399 mm for the west climate zone, middle climate zone, east climate zone, and northeast climate zone during 2003–2020 based on GLEAM (Figure 6b). The net precipitation recharge, which was the difference value between precipitation and evapotranspiration, was conducive to water resources in the northeast climate zone and the east climate zone, and inimical to water resources in the west climate zone and middle climate zone (Figure 6c).

Previous studies on the responses of water resources to climate conditions were mainly conducted in local regions of Inner Mongolia. For example, Chen et al. [69] showed there were no obvious relationships between climate changes and groundwater changes in the middle of Inner Mongolia. Few studies revealed the spatial variations of water storage in the whole of Inner Mongolia. In this study, we studied the responses of regional water storage to climate changes in different climate zones of Inner Mongolia.

On one hand, the responses of water storage to climate changes showed spatial effects. Table 1 showed the decomposition results of TWSA with its hydrological components and precipitation/evapotranspiration in different climate zones. The precipitation and evapotranspiration showed an increasing trend, while the TWSA showed a decreasing trend in the west climate zone, middle climate zone, and east climate zone. In contrast, the TWSA, its hydrological components, and precipitation/evapotranspiration all showed an increasing trend in the northeast climate zone. In addition, the annual phase ($\theta_{ann}$) and semi-annual phase ($\theta_{semi-ann}$) of TWSA were similar with that of precipitation/evapotranspiration in the northeast climate zone and clearly different to that of precipitation and evapotranspiration in the other three climate zones.

On the other hand, the responses of water storage to climate changes showed delay effects, which were identified by cross-correlation analysis. The cross-correlation coefficients between precipitation/evapotranspiration/net precipitation recharge and TWS changes were computed both on the monthly time scale and the annual scale (Figure 7). The TWS changes showed weak correlations with climate factors, with CCF less than 0.5 in all climate zones for the monthly time scale. On the annual scale, TWS showed different degrees of delay to climate factors in different climate zones. TWS in the northeast climate zone showed a high correlation with precipitation, a moderate correlation with evapotranspiration, and a significant positive correlation with net precipitation recharge without lags. One-year shifted TWS changes were positively correlated with precipitation and net precipitation recharge in the east climate zone and middle climate zone. The TWS changes in the west climate zone were more related to evapotranspiration than precipitation and significantly weakly related to net precipitation recharge.

### 4.2. The Influences of Anthropogenic Activities on Terrestrial Water Resources

In addition to climate changes, the influence of anthropogenic activities on water resources cannot be ignored. The continuous increase in water consumption in agricultural irrigation, industrial, and domestic water demand has resulted in the overexploitation of water resources and the aggravation of water resource storage. The water consumption index is the mean water consumption from 2000–2020 divided by the area. The spatial distribution of the water consumption index coincided with that of TWSA in Inner Mongolia (Figure 8).

Inner Mongolia is located in arid and semi-arid regions, and natural precipitation cannot meet the needs of crop growth and development. Taking corn as an example, under natural climates condition, precipitation could not meet the breeding needs of corn, and the crop was essentially in a state of water deficit with only a small surplus in some parts of northeastern Inner Mongolia [54]. Therefore, the irrigation quota was high in Inner Mongolia, and agriculture was the most important user of water resources. The study of Liu et al. [70] showed that climate drought was the major factor for the surface water shortage in Inner Mongolia in the last century, while the water use in agriculture-dominated regions had accelerated the shortage since 2000. According to statistical yearbook data [61], the area of cultivated land increased from 3.967 million ha in 1947 to 9.272 ha in 2018. Correspondingly, the irrigated land increased from 0.295 million ha in 1947 to 2.923 million ha in 2018. Agricultural regions were mainly located in the east climate zone and middle climate zone of Inner Mongolia, where they showed non-obvious responses to climate change. The annual increase in agricultural irrigated area was bound to increase the dependence on water resources under current climate conditions. According to the water resource bulletin [62], the average annual water consumption in Inner Mongolia was 18.2 billion m^3^ from 2000 to 2020, of which agricultural water consumption accounted for 73%.

As a vital source of freshwater, groundwater plays an important role in agriculture in many parts of the world [71,72]. It has been reported that agricultural irrigation caused the over-exploitation of groundwater, resulting in the reduction of groundwater storage in northern China [73,74,75,76]. Figure 8 showed the area equipped for irrigation, which was extracted from the global map of irrigation areas (GMIA) of FAO (http://www.fao.org/, accessed on 13 October 2022). According to the GMIA, the area equipped for irrigation was 3,332,520 ha, with 2,150,064 ha dependent on groundwater. The spatial distribution of the irrigated area coincided well with that of GWSA in Inner Mongolia. Therefore, agricultural irrigation is one of the major driving factors of the reduction in water resources (especially groundwater resources) in the agropastoral transitional zone.

On the other hand, with the development of the social economy, industrial and domestic water consumption showed an increasing trend in recent years. The total energy production increased rapidly to support the huge economic activity expansion in Inner Mongolia. Coal makes up more than 95% of the total energy production in Inner Mongolia. Coal mining has been widely operated across Inner Mongolia, especially in the grassland area [77]. The coal reserves increased from 2.23 × 10^4^ billion tons in 2001 to 5.18 × 10^4^ billion tons in 2020. Coal mining is an extremely water-intensive industry [78], in which 0.5 tons of water is consumed and 4 tons of pit water is drained for every 1–2 tons of coal [79]. Therefore, the large-scale and increasing amounts of coal mining aggravated the water shortage in the middle of Inner Mongolia.

### 4.3. Hydrological Drought Evaluation Based on GRACE

Water resources are the basic natural resources, an important controlling factor of the ecological environment, and also a part of strategic economic resources. However, the water resource deficit is the main characteristic of the climate water balance under natural precipitation and evapotranspiration in Inner Mongolia [54]. The water storage deficit (WSD) based on GRACE data was used to identify the severity of hydrological droughts and quantify the water needed to return to normal conditions. Negative WSD values represent deficits in water storage compared to its monthly mean values, while positive values signify surplus water storage. In this study, a hydrological drought event is defined as when the negative WSD lasts 6 months. The WSD is combined with event durations to identify the severity of a hydrological drought event. Table 2 summarizes the hydrological drought events in different climate zones.

Figure 9a–d showed the time series of WSD in different climate zones. For the northeast climate zone, where the precipitation resources were abundant, six hydrological drought events were identified with the mos severe peak magnitude of 127.64 mm and the longest duration of 30 months. In addition, hydrological droughts coincided with precipitation almost without time lags, indicating climate factors were the major factors for water storage changes.

For the west climate zone, where the precipitation resources were limited, six hydrological drought events were also identified with the most severe peak magnitude of 52.18 mm and the longest duration of 38 months. The hydrological droughts also coincided with precipitation but with a several-month lag. Compared with other regions, the decline rate of water storage was minimal in this region, and the severity of hydrological droughts was the lowest. However, the water deficits were continuously serious, especially after the 2010s. It should be considered that the precipitation in this region is low, and the potential evapotranspiration is high. Once the water storage continues to decline, it is difficult to recover even though the amount is small.

For the middle climate zone, four hydrological drought events were identified with the severest peak magnitude of 73.29 mm and the largest duration of 48 months. For the east climate zone, five hydrological drought events were identified with the most severe peak magnitude of 109.21 mm and the longest duration of 35 months. It was worth noting that WSD did not always coincide with precipitation in these two climate regions, especially after 2015. In fact, the hydrological droughts based on GRACE did not always show good conformity with meteorological droughts. The study of Tapley et al. [80] found that GRACE-derived hydrological droughts had longer drought persistence, relative to that based only on meteorological variables. This is because that hydrological droughts based on WSD capture both the anthropogenic effects and natural drought responses. The inconsistency between climate droughts and hydrological droughts in the east and middle climate zones illustrated the anthropogenic activities that played a major role in the water resource changes.

## 5. Conclusions

Monitoring the terrestrial water storage (TWS) changes in Inner Mongolia are crucial for water resource rational utilization and ecological restoration, in which there are serious water resource shortages. Based on the results, the major conclusions are as follows:(1)Water shortage has become a bottleneck restricting regional development in Inner Mongolia. During the period of 2003–2021, the TWSA decreased at the rate of 1.82 mm/yr based on GRACE data, the SMA increased at the rate of 2.15 mm/yr, and the SWEA exhibited no significant trend based on LSMs. Correspondingly, the GWSA decreased at the rate of 4.15 mm/yr based on the water balance.(2)The responses of water storage changes to climate change were studied in different climate zones of Inner Mongolia by time-series analysis. The TWSA increased at the rate of 2.36 mm/yr during the period of 2003–2021 for the northeast region due to the abundant precipitation.(3)Compared with climate change, human activities played a dominant role in the water storage shortage of the middle and eastern parts of Inner Mongolia. The TWSA in the east and middle climate zones evidently decreased at the rates of 4.09 mm/yr and 3.69 mm/yr due to the overexploitation of groundwater for agricultural irrigation and coal mining.(4)For western regions, less precipitation and more evapotranspiration were not conducive to the replenishment of water resources. However, the terrestrial water resources in this region did not change significantly due to stable meteorological conditions and fewer human activities.(5)The water storage deficits were the most severely affected by both climate conditions and human activities. With the surging population and increasing demand for food, the amount of agricultural irrigation will continuously increase. Future risks of water resource deficits will likely rise due to the increasing reliance on groundwater to fulfill water demands.

## Figures and Tables

**Figure 1 sensors-22-09665-f001:**
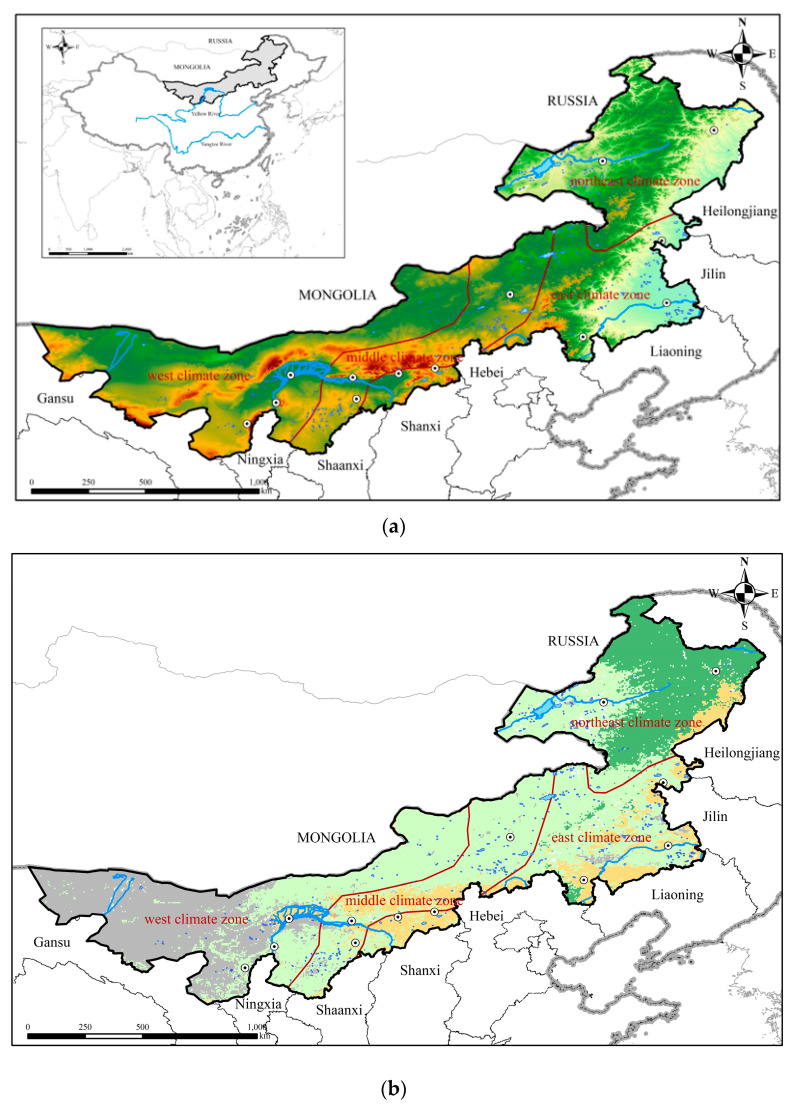
Location (**a**) and land cover (**b**) of Inner Mongolia.

**Figure 2 sensors-22-09665-f002:**
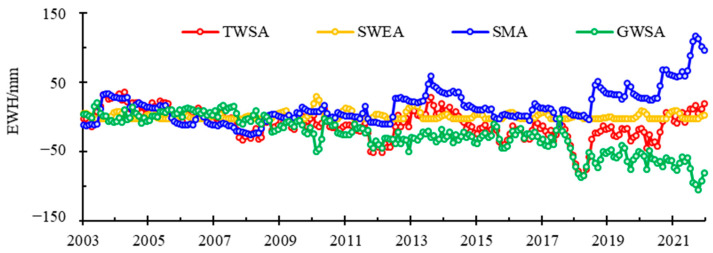
Temporal changes of terrestrial water storage and its hydrological components.

**Figure 3 sensors-22-09665-f003:**
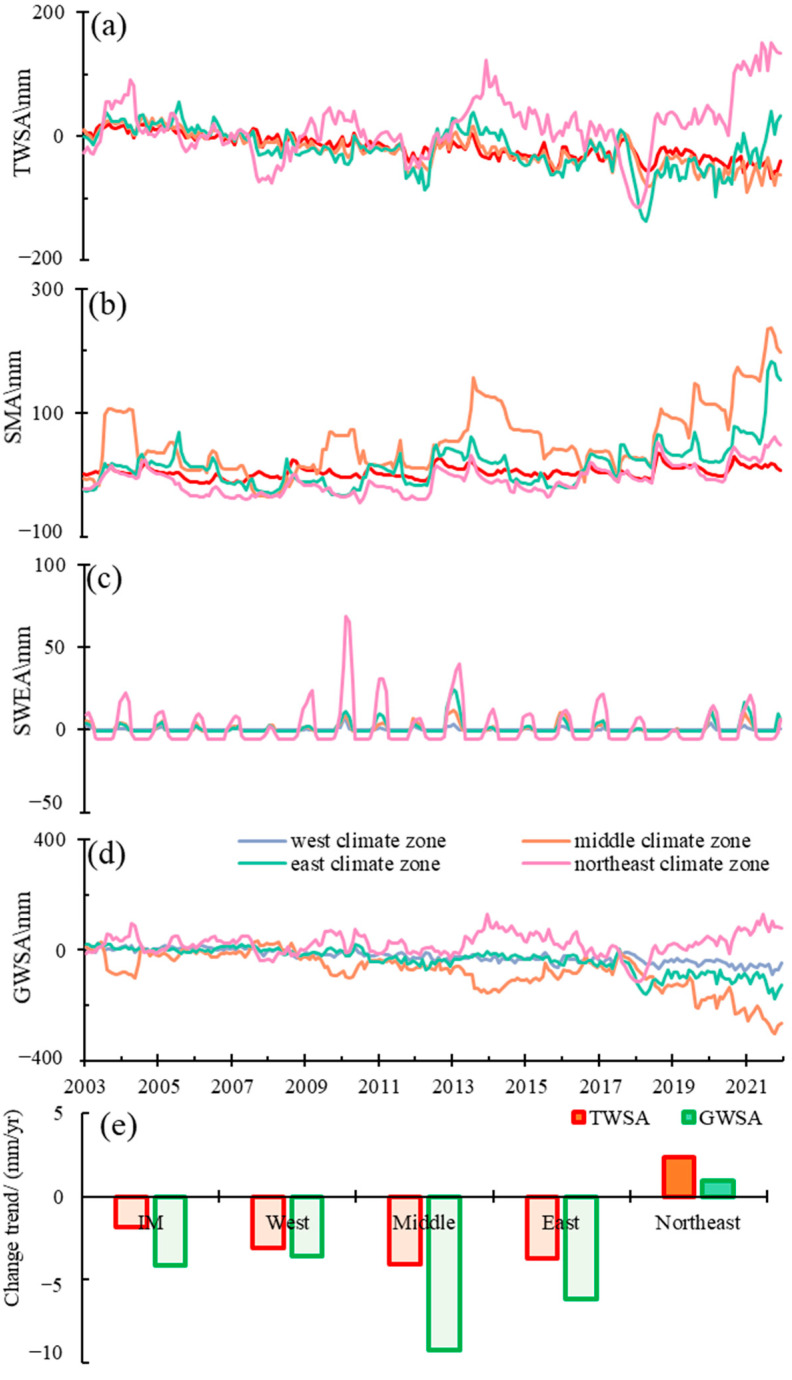
Temporal changes of TWSA and its components (SMA, SWEA and GWSA) in different climate zones. ((**a**) TWSA from CSR-M, (**b**) SMA from GLDAS_NOAH, (**c**) SWEA from GLDAS_NOAH, CLSM, and VIC, (**d**) GWSA based on water balance theory, (**e**) change trends of TWSA and GWSA in different climate zones (IM: Inner Mongolia; West: West climate zone; Middle: Middle climate zone; East: East climate zone; Northeast: Northeast climate zone)).

**Figure 4 sensors-22-09665-f004:**
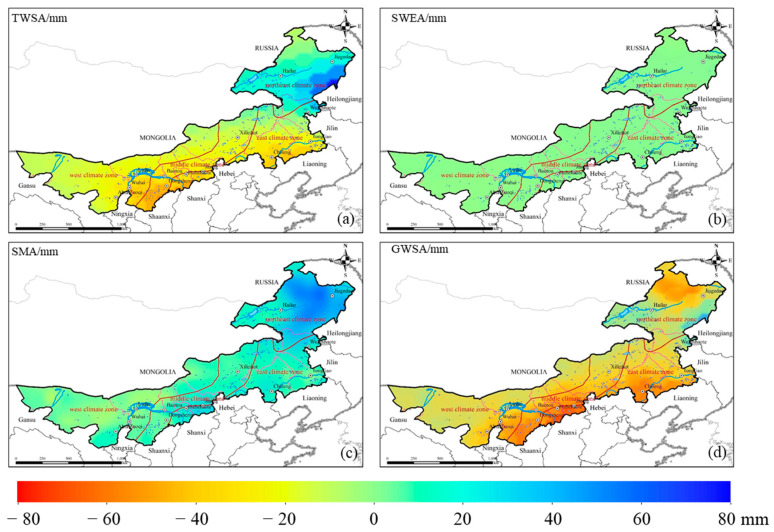
Spatial distribution of mean annual TWSA and its components (SMA, SWEA, GWSA) in 2003–2021.

**Figure 5 sensors-22-09665-f005:**
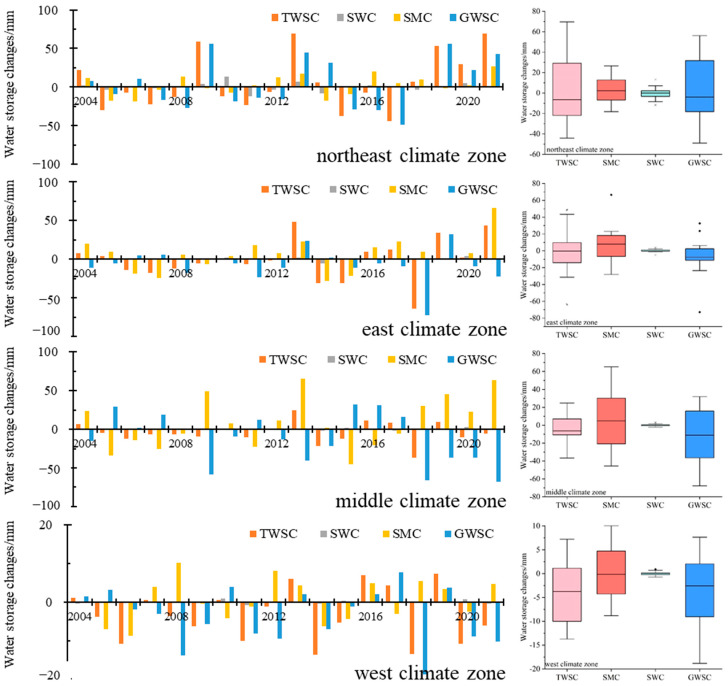
Time series and box diagram of TWS changes with its hydrological components (TWSC: TWS changes; SMC: Soil moisture changes; SWC: Surface water changes, the sum of SWE and CanoInt; GWSC: Groundwater storage changes). (The asterisk in the box diagram is the outlier).

**Figure 6 sensors-22-09665-f006:**
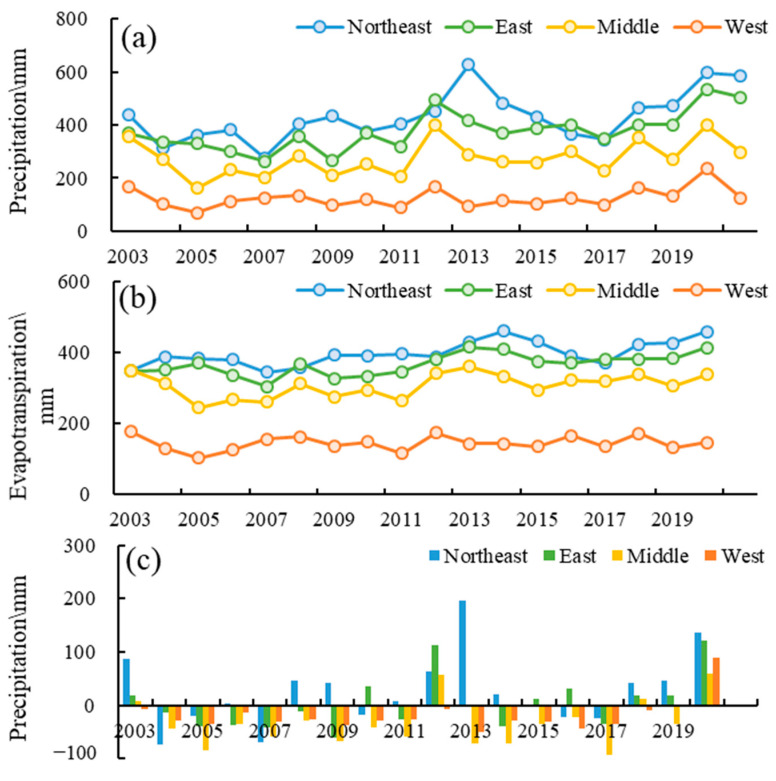
The climate conditions in Inner Mononlia ((**a**) precipitation in 2003–2021, (**b**) evapotranspiration in 2003–2020, (**c**) net precipitation recharge in 2003–2020 (Northeast: Northeast climate zone; East: East climate zone; Middle: Middle climate zone; West: East climate zone;)).

**Figure 7 sensors-22-09665-f007:**
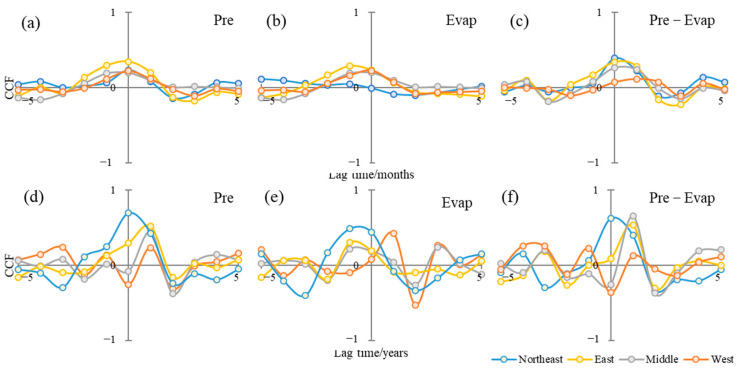
The cross-correlation coefficients (CCF) between precipitation (**a**), evapotranspiration (**b**) and precipitation minus evapotranspiration (**c**) with TWS changes on monthly and precipitation (**d**), evapotranspiration (**e**), and precipitation minus evapotranspiration (**f**) with TWS changes on annual time scales (Pre: Precipitation as input; Evap: Evapotranspiration as input; Pre–Evap: Precipitation minus evapotranspiration as input).

**Figure 8 sensors-22-09665-f008:**
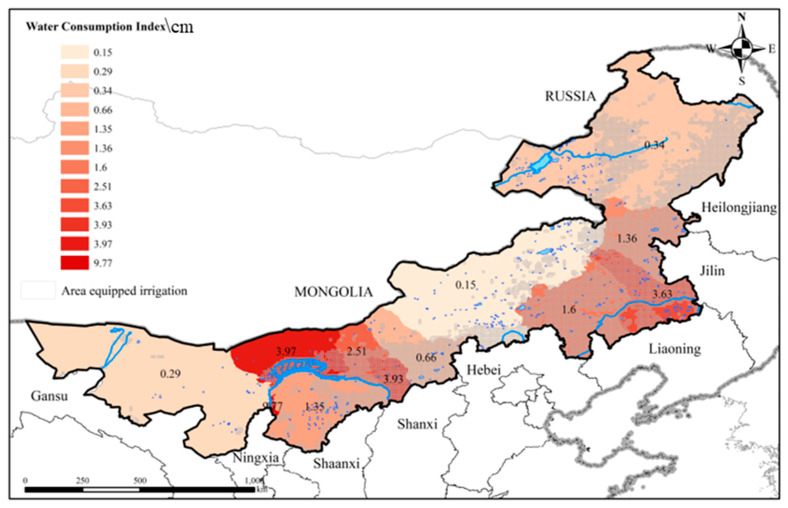
Distribution map of irrigated area and water consumption in Inner Mongolia.

**Figure 9 sensors-22-09665-f009:**
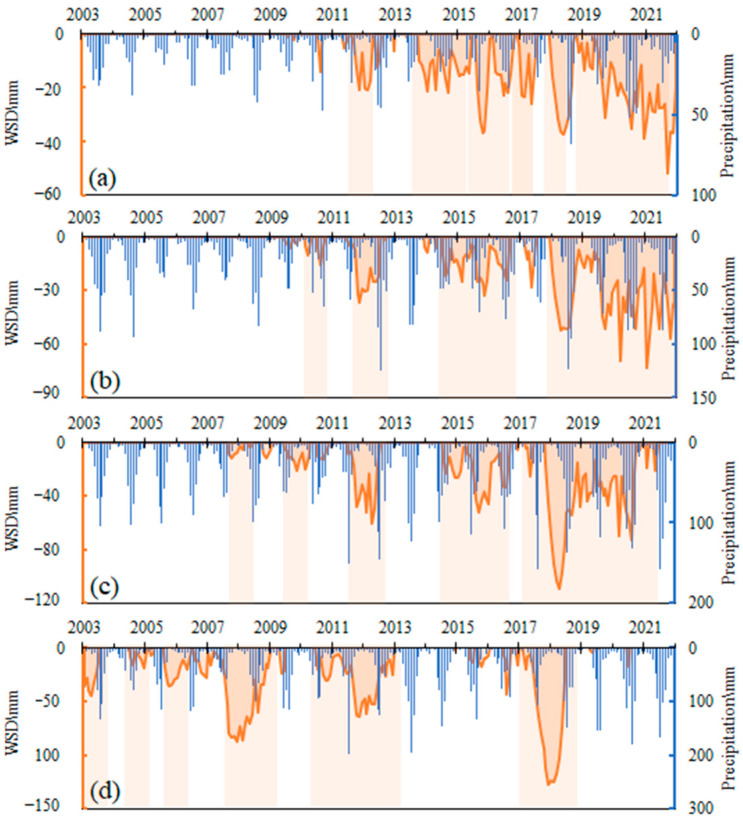
Water storage deficit based on GRACE in different climate zones (Blue histogram: Precipitation, orange shaded areas: Water storage deficits (WSD)).

**Table 1 sensors-22-09665-t001:** Decomposition of TWSA and its hydrological components with precipitation and evapotranspiration.

Variable	*b*	$A_{ann}$	$\theta_{ann}$	$A_{semi-ann,}$	$\theta_{semi-ann,}$	*R* ^2^
West Climate Zone						
Precipitation	0.02	11.81	0.91	4.34	−0.10	0.52
Evapotranspiration	0.00	12.49	0.94	4.33	−0.37	0.83
TWSA	−0.26	1.84	0.25	2.43	−0.45	0.74
SMA	0.05	5.39	−0.05	2.01	1.51	0.31
GWSA	0.30	3.48	−0.41	2.00	0.37	0.80
Middle Climate Zone						
Precipitation	0.03	28.25	0.90	11.46	0.19	0.71
Evapotranspiration	0.02	27.16	1.08	7.07	−0.20	0.93
TWSA	−0.33	4.26	−0.37	4.63	−0.37	0.69
SMA	0.46	7.60	−0.66	4.53	1.11	0.33
GWSA	−0.80	5.46	−1.46	5.40	0.35	0.58
East climate zone						
Precipitation	0.06	41.34	0.99	18.32	0.43	0.75
Evapotranspiration	0.03	33.25	1.10	8.28	−0.09	0.97
TWSA	−0.29	10.07	0.00	5.35	−0.07	0.36
SMA	0.27	11.33	0.00	4.08	−0.64	0.31
GWSA	−0.57	2.83	1.43	1.44	1.11	0.73
Northeast Climate Zone						
Precipitation	0.07	45.61	1.01	22.54	0.38	0.72
Evapotranspiration	0.03	39.18	1.23	6.67	0.58	0.97
TWSA	0.22	3.03	1.35	0.84	0.39	0.09
SMA	0.17	6.90	0.19	2.13	−1.48	0.31
GWSA	0.05	12.21	1.41	5.27	−0.70	0.06

Notes: b is long-term trend of time series, $A_{ann}$ and $A_{semi-ann}$ are the annual and semi-annual amplitudes after being revised by desmoothing, $\theta_{ann}$ and $\theta_{semi-ann}$ are the annual and semi-annual phases, and *R*^2^ is the coefficient of the time series decomposition regression function.

**Table 2 sensors-22-09665-t002:** Summary table of hydrological drought events identified by GRACE.

Climate Zone	Num	Time Span (Year.m)	$D\left(t\right)$ (Month)	Se (mm)	Mean (mm)	Peak Magnitude (mm)
West climate zone	6	2011.8–2012.3	8	102.86	12.86	20.98
		2013.10–2015.5	20	258.38	12.92	21.93
		2015.8–2016.10	15	273.71	18.25	37.06
		2016.12–2017.7	8	113.1	14.14	25.96
		2017.12–2018.9	10	233.35	23.34	37.34
		2018.11–2021.12	38	801.39	21.09	52.18
Middle climate zone	4	2009.6–2009.11	6	20.57	3.43	6.57
		2011.6–2012.7	14	271.75	19.41	36.99
		2014.7–2016.8	26	410.77	15.80	33.21
		2018.1–2021.12	48	1648.01	34.33	73.29
East climate zone	5	2007.9–2008.2	6	44.47	7.41	11.37
		2009.6–2010.4	11	135.59	12.33	21.53
		2011.7–2012.7	13	396.97	30.54	60.08
		2014.7–2016.9	27	618.44	22.91	51.73
		2017.11–2020.9	35	1659.37	47.41	109.21
Northeast climate zone	6	2003.1–2003.7	7	207.9	29.70	45.12
		2004.7–2005.1	7	65.04	9.29	18.25
		2005.8–2006.6	11	252.3	22.94	35.91
		2006.8–2009.1	30	1183.76	39.46	87.24
		2010.7–2012.12	30	876.85	29.23	63.97
		2017.2–2018.6	17	1151.88	67.76	127.64

Num is the hydrological drought event number; $D\left(t\right)$ is the duration of hydrological drought event; Se is the water storage deficit severity; Mean: The mean water deficit in hydrological drought event; Peak: The peak magnitude of hydrological drought event.

## Data Availability

The GRACE and GRACE Follow-On data were collected from http://www2.csr.utexas.edu/GRACE/RL06_mascons.html, accessed on 13 October 2022). The three land surface models were collected from http://disc.sci.gsfc.nasa.gov/serVICes/grads-gds/gldas/, accessed on 13 October 2022). The land use/cover map was from https://www.resdc.cn/data.aspx?DATAID=335, accessed on 13 October 2022). The gridded precipitation data were obtained from Global Precipitation Measurement (GPM) (https://gpm.com.hk/, accessed on 13 October 2022). The evapotranspiration data were from Global Land Evaporation Amsterdam Model (http://www.gleam.eu, accessed on 13 October 2022). The human water consumption data including agricultural consumption were obtained from water resources bulletins (http://slt.nmg.gov.cn/xxgk/bmxxgk/gbxx/szygb/, accessed on 13 October 2022) or statistical yearbooks (http://tj.nmg.gov.cn/tjyw/jpsj/index_1.html, accessed on 13 October 2022).

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
