# Peer review of "Evaluation of Terrestrial Water Storage Changes and Major Driving Factors Analysis in Inner Mongolia, China"

_sensors, 2022, doi:10.3390/s22249665_

Round 1

Reviewer 1 Report

The comments for the manuscript were uploaded to the system as a PDF file.

Author Response

Response to the review report

Title: Evaluation of terrestrial water storage changes and major driving factors analysis in Inner Mongolia, China

Journal-and Manuscript ID: sensors-1997476  

Respected reviewer:

Before responding to your comments, please allow us to express our sincerest gratitude for your carefully reviewing this paper and for your objective comments. It is no exaggeration to say that your suggestions have milestone significance not only for this manuscript but also for the authors’ future research. We have studied your comments carefully and respond to them one by one in following, hoping that you will not be disappointed for the revised version.

  1. The Introduction Section needs to be revised, the contribution of cited literature should be addressed in a proper scenario relevant to the article’s topic. The previously published studies that investigate the water storage variations exactly in the studied area should be cited and discussed. The weaknesses and drawbacks of the GRACE/GRACE-FO missions, land surface models in hydrology research should be mentioned with addressing if these weaknesses affected this study or not (especially considering the size of the investigated basins and significance of the change in hydrology signal in this study).

Thank you for the comments and suggestions.

Firstly, we check and modify some language expressions and grammatical tenses of the Introduction Section.

Secondly, we add some references, including the previous studies on water resources investigation in the study area. For detail, please check: Page 2, line 71-74:

“The deficit of water resources has become an important factor for the restoration of ecological environment in Inner Mongolia. Previous studied have preliminarily ex-plored the water resources statue and supply/demand balance [51-54]. However, there is a lack of studies on the regional water resources changes in Inner Mongolia and re-sponses to climate changes and human activities.”

Finally, we add the discussion of the weaknesses of GRACE/GRACE-FO missions and land surface models in hydrology research and its influences on this study. For detail, please check: Page 2, line 59-67:

“Despite the above advantages, it cannot be denied there are limitations to study water resources changes by GRACE and LSMs. The coarse spatial resolution of GRACE data is the primary factor affecting the TWS changes at basin scale [44,45]. LSMs also have certain limitations in simulating the soil moisture in deep soil layers [46]. However, the study of Swenson et al.[47] showed that if the study area reached more than 40×104 km2, the error of TWS changes based on GRACE data would be less than 1 cm, and the larger the study area, the higher the accuracy. In addition, recent studies have demonstrated that water storage changes inferred from the GRACE data and LSMs have sufficient resolution for regional water resources management [25,48-50].”

  1. Line 43: “the effects of large earthquakes” (how large, please states as greater than … magnitudes etc.”

Thank you for this suggestion. We refer to relevant literatures and rewrite the sentence as “the effects of earthquakes down to magnitude 8.3 on earth gravity”. For detail, please check in Page 1 line 41.

  1. Line 47: (GRACE also includes GRACE Follow On, hereafter) -> it is not clear, please express more clearly, you can say “GRACE terminology will be used for both GRACE and GRACE-FO missions in the article”.

Thank you for your advice, our original expression (GRACE also includes GRACE Follow On, hereafter) does have unclear meaning, but we have not found a native English expression. Thank you for your reference, which we think is very professional and academic. Therefore we have replaced this sentence with the original sentence on Page 2 line 45-46.

  1. Lines 49-50:”First, the GRACE based TWS ...” sentence should be expressed more clearly.

Thanks for this comment. We rewrite this sentence as “Firstly, the GRACE-based TWS changes are the integrated the changes in all forms of water [8]. Therefore, GRACE data could be used to quantify groundwater storage (GWS) changes [26,27], runoff discharge [28-30], and evapotranspiration [31-33] with the assistant of land surface models (LSMs). Secondly, GRACE-based TWS changes capture the comprehensive responses to both natural environmental changes and an-thropogenic activities [34]. For some regions, climate changes have been regarded as the dominant factors controlling the water storage [35,36]. While in other regions, the direct and indirect consequences of anthropogenic activities can also be linked to water shortage decreased [37,38]. In addition, GRACE data extends a new way to reflect the water deficit and to provide additional information about the effects of meteorological drought, and quantify the severity of hydrological drought [39-43]”. For detail, please check in Page 2, line 48-58.

  1. Line 74: clarify what “its major driving factors” means, it should be told explicitly.

Thanks for this comment. In fact, the major driving factors for the water resources changes in Inner Mongolia include the climate changes (precipitation and evaporation) and human activities (agricultural irrigation and coal mining). We rewrite this sentence as “This study could provide a basic framework for water resources changes in Inner Mongolia and the responses of its water resources changes to climate changes and human activities, which are beneficial to water resources management and ecological restoration”. For detail, please check in Page 2, line 80-83.

  1. Line 83:”The terrain is sloping…” instead use “Topography has a slope…”

Thanks for this suggestion. We rewrite this sentence following you suggestion as “Topography has a slope from northeast to southwest…”. For detail, please check in Page2, line 91.

  1. In the methodology: All the formulas should be re-considered and written properly, because the symbols, indices were not written properly, on the other hand each parameter in each formula should be introduced and explained carefully. See e.g. The Mann-Kendall trend test, you introduce Xt, Yt, but the formula includes xi, xj, what is I, what is j? Similar confusing situation is valid for some other formulas as well. So methodology part should be revised extensively.

Thanks for this comment. The methodology part have been revised as following:

  • We modify the explanation of Mann-Kendall trend test. For detail, please check Page 6, line 170.
  • We modify the equations (5) and (6), replacing clmate_factor with x, and TWS changes with y respectively. And we supplement the meaning of each variable in the equation in the text. For details, please check in Page 6, line 176-179.
  • The explanation of ( ) and  (  are added in Page 6, line 178.

  1. 6: how do you calculate sigma(x) and sigma(y), there is no explanation?

Thanks for this suggestion. As mentioned in the previous response, we add an explanation to sigma(x) and sigma(y) in Page 6, line 178.

  1. In section 2.3.5: why do you consider only 12 month and 6 month periods, further explanations and citing references are expected?

Thanks for this question and suggestion.

TWS changes derived from GRACE data include long trend changes and seasonal variations. As far as the author knows, there are mainly two methods for TWS decomposition. One is to divide the time series into “trend + seasonal variation + residual” based on STL method; another is to decompose the time series into long-term trend and seasonal change signals according to reference (doi:10.3390/w12113128). One advantage of second method is that the seasonal signal provides the annual terms and semi-annual terms. The amplitude and phase of the annual and semi annual of TWS changes time series can be compared with that of climate factors (precipitation and evaporation) to study the responses of water resources to climate changes.

Following you suggestion, we added two references. For detail, please check on page 6, line 182, and line 186-189.

  1. Please do not use” while stating a time interval, prefer”-“for example “2003-2021”.

Thanks for the suggestion. We replace “~” with “–”in the whole text.

  1. Figure 1: please keep the maximum and minimum range of vertical axes the same for each graphic.

Thanks for the suggestion. We have reworked Figure 1(2?) to ensure that the range of vertical axes is consistent. For detail, please check in Page 7 , line 198.

  1. Line 170: Figure 2 was cited, but Figure 2 is too far from this address.

Thank you for this comments. We add the cite of Figure 2(3?) in Page 7, line 226, so the figure at line 229 is not too far away.

  1. In the article there are two Figures with the number 4 (?)

Thank you for pointing out the error in the position and serial number of the figures or tables. We check and modify the figure positions of the full text and the cite positions relative to the figures.

  1. In Figure 3 (page 11): why the temperature graphic covers the years between 1948 and 2018? Whereas the others cover between 2003-2020 years.

Thanks for this question.

Actually, there are two main reasons for the longer period in temperature. First, it makes sense to talk about climate changes only if you know whether an area is warming or cooling. However, the air temperature changes are long-term, and the absolute temperature change is relatively small. According to the IPCC published data, the globally surface temperature showed a warming of 0.85 °C over the period 1880 to 2012. It is difficult to define the change over a decade as a trend or a normal fluctuation. Second, the changes of precipitation and evapotranspiration often lag the changes of temperature. However, the lag of TWS on evapotranspiration of precipitation is obviously shorter than that of precipitation and evapotranspiration on temperature. For the above two reasons, the time series of temperature is longer.

While there are reasons to choose temperature data over a longer period of time, your question makes us realize that there is something strange about this figure structure. So we reworked Figure 3 to keep the time frame consistent. For detail, please check in Page11, line277.

  1. Is there any unit for “water consumption index”?

Thank you for your question. The unit is cm. For detail, please check in Page 14, line 356.

Finally, thank you again for reviewing the manuscript.

Best wishes from all authors

Reviewer 2 Report

Main review:

------------

The study provides a pretty interesting assessment of TWS variations, and an investigation on the corresponding driving factors for Inner Mongolia of China. The methodology and the used data are basically in accordance with standards. The conclusions are reasonable, no unreasonable statement is done. All in all, it is a well-founded and high level investigation, with no particular weekness in it. There are some minor points, which are suggested to be improved. Here they are:

(1) The reference list is incomplete. Also, the way of referencing in the text is not cosequent. Some suggestions are provided in the "detailed review" part of this review.

(2) It would be worth to indicate differences of this study and Thomas et al [44] in a separated paragraph in the Introduction chapter.

(3) For averaged periodic signals the amplitude is smoothed, the desmoothing of DOI:10.1093/gji/ggv092 should be applied. If the authors disagree with the need of desmoothing, for the sake of completeness, this issue at least should be discussed in the discussion section.

(4) The conclusions listed in the Conclusions chapter are fair. Nevertheless, much more conclusions could be drawn. I suggest to elaborate the conclusions chapter more.

Detailed review:

----------------

Page 2, line 47: some references for GRACE-FO should be listed, e.g. 

DOI:10.1029/2020GL088306

It makes sense to include GRACE-FO references in the context of water storage changes, e.g.

DOI:10.1029/2021JB022124

Page 2, lines 77-92: for the introduction of Inner Mongolia, also some references should be included. For any information, which is not the result of the work of the authors, references to indicate the source of information are essential.

Page 3, line 105: the provided link is not accessible. Instead of providing a link, it would make more sense to provide a reference to the data as the location of the corresponding files may anytime change in the future. Such a reference can be 

DOI:10.1002/2016jb013007

The use of a reference rather than a link is general comment for each data sources, e.g. GLDAS, GPM, GLEAM. Of course, when only online source is available, then it can be used, but it is better to include it in the reference list than in the text.

Page 4, line 121: "CanoInt values from NOAH were significantly higher than those of other two models." - It would be important to understand the reason of the difference of the data sources, why NOAH data is that much different from CLSM and VIC. It might be that CLSM and VIC are not independent, so their similarity does not justify its preference over NOAH.

Page 4, line 149: a reference to the Mann-Kendall test would be welcome, as it is not part of the basic mathematical knowledge, e.g.

Goswami, B. (2017): Mann-Kendall Test, available at https://up-rs-esp.github.io/mkt/

Page 5, equations (5) and (6): You use variables "clmate_factor" and "TWS", but their standard deviation is labelled as "x" and "y". Be consequent! Actually, the use of variable "clmate_factor" is rather messy, I suggest to change the name of this variable, and explain its content in the text.

Page 5, equation (6): as this equation normalizes the cross-correlation, I would rather call the left-hadn side variable as PCC (Pearson correlation coefficient) or NCC (normalized cross-correlation)

Page 5, equation (7): replace "f os" to "f cos".

Page 5, equations (7), (8) and (9): When you use monthly averaged data, it is smoothed due to the averaging. Accordingly, annual and semi-annual amplitudes determined by equation (7) is underestimates the actual amplitude by 1.15% (annual) and 4.72% (semi-annual). See 

DOI:10.1093/gji/ggv092

The desmoothing can be applied for different forms of time series equations containing long-term trend and periodic seasonal components.

The consequence of averaging can be easily restored by multiplying the estimated annual amplitude in (8) by 1.0115 and the semi-annual amplitude in (9) by 1.0472, so it is worth to apply. Note that it does not impact the estimated linear trends but the amplitudes only, e.g. values in lines 202 and 203, in Table 1, 

Page 10, line 248: insert a reference for the NCEP-NCAR reanalysis data, typically it is

DOI:10.1175/1520-0477(1996)077<0437:TNYRP>2.0.CO;2

Page 12, lines 289 and 290: The introduction of the variables of Table 1 in this for is incomplete. Please form a full sentence, and refer to Table 1 in the text.

Page 13, lines 337 and 338: "According to the Inner Mongolia Water Resources Bulletin" - again incorrect referencing. Include the reference in the reference list and refer in the text with its reference [number].

Author Response

Response to the review report

Title: Evaluation of terrestrial water storage changes and major driving factors analysis in Inner Mongolia, China

Journal-and Manuscript ID: sensors-1997476  

Respected reviewer:

Before responding to your comments, please allow us to express our sincerest gratitude for your carefully reviewing this paper. The comments are worth learning, not only meaningful for this study, but also for future research. In the following part, the authors respond to your detailed reviews one by one, hoping that you will not be disappointed for the revised version.

Q1: Page 2, line 47: some references for GRACE-FO should be listed, e.g. DOI: 10.1029/2020GL088306. It makes sense to include GRACE-FO references in the context of water storage changes, e.g.DOI:10.1029/2021JB022124

A: Thank you for your advice. References about GRACE-FO really need to be quoted. The two articles you recommended are highly representative and scientific in the fields related to GRACE-FO and water storage changes. Therefore we refer them in Page2, line 45.

Q2: Page 2, lines 77-92: for the introduction of Inner Mongolia, also some references should be included. For any information, which is not the result of the work of the authors, references to indicate the source of information are essential.

A: Thanks for this comment. We greatly respect your starting point of respecting intellectual property rights and protecting the rights and interests of data providers. We listed three references in Page 2, line 89, line 91, and Page 3, line 96, respectively. In addition, the source of climatic zones data is attached in Page 3, line 100-101.

Q3: Page 3, line 105: the provided link is not accessible. Instead of providing a link, it would make more sense to provide a reference to the data as the location of the corresponding files may anytime change in the future. Such a reference can be DOI:10.1002/2016jb013007. The use of a reference rather than a link is general comment for each data sources, e.g. GLDAS, GPM, GLEAM. Of course, when only online source is available, then it can be used, but it is better to include it in the reference list than in the text.

A: Thanks for your suggestion. This suggestion is not only beneficial for this study, but also helpful for future paper writing. For the data used, we list their corresponding references in right locations, which can be seen in Page 4, line 124 for GRACE/GFO data, Page 5, line 130 for GLDAS, Page 5, line 136 for GPM and Page 5, line 138 for GLEAM, Page 5, line 140 for statistics data, respectively. In addition, we express our gratitude to the data providers and place the currently available links in the part of “Acknowledgments”.

Q4: Page 4, line 121: "CanoInt values from NOAH were significantly higher than those of other two models." - It would be important to understand the reason of the difference of the data sources, why NOAH data is that much different from CLSM and VIC. It might be that CLSM and VIC are not independent, so their similarity does not justify its preference over NOAH.

A: Thanks for this question. That's a good question, but it's also really hard to answer. Given that the authors did not express the differences clearly, the following figure shows the CanInt values of different climate zones in Inner Mongolia provided by the three land surface models.

It can be seen from the figure that different land surface models have good consistency from 2000 to 2020, while the CanInt values in 2021 provided by NOAH is significantly higher than the other two. And the magnitude of the CanInt values in 2021 provided by NOAH is questionable.

Regarding the inconsistency of data provided by NOAH in 2021, we have two points to make. Firstly, the data is processed in batches, so there is no problem of processing method. Secondly, the value of other parameters in 2021 provided by NOAH do not appear inconsistent phenomenon.

Therefore, considering that the algorithm of the land surface model itself is not the focus of this study, considering the consistency and magnitude reliability of the data provided by the other two models, we choose the mean value of CLSM and VIC. Of course, we will pay close attention to the latest version of the data.

Q5: Page 4, line 149: a reference to the Mann-Kendall test would be welcome, as it is not part of the basic mathematical knowledge, e.g.Goswami, B. (2017): Mann-Kendall Test, available at https://up-rs-esp.github.io/mkt/

A: Thanks for this suggestion. According to your suggestion, we have added two references on Man-Kendall trend test. For detail, please check Page 5, line 169. We calculate the trend of time series based on Matlab. But the information about the python you provided is a great help to us.

Q6: Page 5, equations (5) and (6): You use variables "clmate_factor" and "TWS", but their standard deviation is labelled as "x" and "y". Be consequent! Actually, the use of variable "clmate_factor" is rather messy, I suggest to change the name of this variable, and explain its content in the text.

A: Very thanks for this comment. Equations (5) and (6) are really confusing and unclear. In order to be clearer, we modify the equations (5) and (6), replacing clmate_factor with x, and TWS changes with y respectively. And we supplement the meaning of each variable in the equation in the text. For details, please check in Page 6, line 176.

Q7: Page 5, equation (6): as this equation normalizes the cross-correlation, I would rather call the left-hadn side variable as PCC (Pearson correlation coefficient) or NCC (normalized cross-correlation)

A: Thanks for this suggestion. The cross-correlation function is defined by Mangin (Alain Mangin. Contribution à l’étude hydrodynamique des aquifères karstiques : Première partie : Généralités sur le karst et les lois d’écoulement utilisées (Ann. Spéléol., 1974, 29, 3, p.283-332). We refer to the literature (DOI:10.1016/S0022-1694(97)00155-8) for the calculation of this parameter.

The Pearson correlation coefficient is used to measure the linear correlation between two variables, X and Y. The cross-correlation coefficient is used to reflect the correlation between X and Y at the time lags of k order. The Matlab examples could be helpful at https://ww2.mathworks.cn/help/matlab/ref/xcorr.html#responsive_offcanvas.

Q8: Page 5, equation (7): replace "f os" to "f cos".

A: Thank you very much for pointing out the error. We are very sorry for this careless printing and it has been corrected.

Q9: Page 5, equations (7), (8) and (9): When you use monthly averaged data, it is smoothed due to the averaging. Accordingly, annual and semi-annual amplitudes determined by equation (7) is underestimates the actual amplitude by 1.15% (annual) and 4.72% (semi-annual). See DOI:10.1093/gji/ggv092. The desmoothing can be applied for different forms of time series equations containing long-term trend and periodic seasonal components. The consequence of averaging can be easily restored by multiplying the estimated annual amplitude in (8) by 1.0115 and the semi-annual amplitude in (9) by 1.0472, so it is worth to apply. Note that it does not impact the estimated linear trends but the amplitudes only, e.g. values in lines 202 and 203, in Table 1.

A: Thank you for this suggestion and discussion. To be honest, we did not pay attention to this article, nor did we think of the data desmoothing. We have studied the article you recommended carefully. But it has to be admitted that, given that this article is of theoretical aspect, we can only understand the general. Thanks again for your explanation in the detailed review, we have a further understanding. The desmoothing is really worth trying and applying. We calculated the latest amplitudes by the method you advised, the desmoothing ampiltudes do not affect the overall conclusions. The specific modifications could be check in Page 6, line 186-189, and Page 12, table 1.

Q10: Page 10, line 248: insert a reference for the NCEP-NCAR reanalysis data, typically it is DOI: 10.1175/1520-0477(1996)077<0437: TNYRP>2.0.CO;2

A: Thanks for this suggestion. According to your suggestion, we have added the references for the NCEP-NCAR reanalysis data. . For detail, please check Page 10, line 256.

Q11: Page 12, lines 289 and 290: The introduction of the variables of Table 1 in this for is incomplete. Please form a full sentence, and refer to Table 1 in the text.

A: Thanks for this suggestion. We rewritten the notes for the Table and we refer the Table 1 in the Page 11, line 288.

Q12: Page 13, lines 337 and 338: "According to the Inner Mongolia Water Resources Bulletin" - again incorrect referencing. Include the reference in the reference list and refer in the text with its reference [number].

A: Thank you for this reminding. We have added the corresponding reference to the Water Resources Bulletin in the text with reference number and revised it accordingly in the reference list. For detail. Please check in Page 13, line337, and line 343.

Finally, thank you again for reviewing the manuscript.

Best wishes from all authors

Reviewer 3 Report

Dear All,

 In this paper, Evaluation of terrestrial water storage changes and major driving factors analysis in Inner Mongolia, China have been conducted. The paper's findings have important implications. However, there are some remarks:

 The abstract need to be corrected for some grammatical errors.

Please improve the Introduction section:

Here are some papers you can use them

Groundwater depletion in the Middle East from GRACE with implications for transboundary water management in the Tigris–Euphrates–western Iran region. Water Resour. Res. 2013, 49, 904–914.

Integrated Geophysical Assessment of Groundwater Potential in Southwestern Saudi Arabia, Frontiers in Earth Sciences, 10.3389/feart.2022.937402.

Please add some geological backgrounds to the Study area section.

Why did not use other sources of GRACE mascon data such as JPL, and GFCF.

Figure 1. Could you please adjust the vertical scale to let us see the variations in the time series?  

Please improve the discussion section.

Author Response

Response to the review report

Title: Evaluation of terrestrial water storage changes and major driving factors analysis in Inner Mongolia, China

Journal-and Manuscript ID: sensors-1997476  

Respected reviewer:

Before responding to your comments, please allow us to express our sincerest gratitude for your carefully reviewing this manuscript. We have studied your comments carefully and respond to them one by one.

  1. The abstract need to be corrected for some grammatical errors.

A: Thanks for this comments. We check and modify some language expressions and grammatical tenses.

  1. Please improve the Introduction section.

Thanks for this comments. We have made the following changes to the introduction:

Firstly, we check and modify some language expressions and grammatical tenses of the Introduction Section.

Secondly, we add some references, including the previous studies on water resources investigation in the study area. For detail, please check: Page 2, line 71-74:

“The deficit of water resources has become an important factor for the restoration of ecological environment in Inner Mongolia. Previous studied have preliminarily ex-plored the water resources statue and supply/demand balance [51-54]. However, there is a lack of studies on the regional water resources changes in Inner Mongolia and re-sponses to climate changes and human activities.”

Finally, we add the discussion of the weaknesses of GRACE/GRACE-FO missions and land surface models in hydrology research and its influences on this study. For detail, please check: Page 2, line 59-67:

“Despite the above advantages, it cannot be denied there are limitations to study water resources changes by GRACE and LSMs. The coarse spatial resolution of GRACE data is the primary factor affecting the TWS changes at basin scale [44,45]. LSMs also have certain limitations in simulating the soil moisture in deep soil layers [46]. However, the study of Swenson et al.[47] showed that if the study area reached more than 40×104 km2, the error of TWS changes based on GRACE data would be less than 1 cm, and the larger the study area, the higher the accuracy. In addition, recent studies have demonstrated that water storage changes inferred from the GRACE data and LSMs have sufficient resolution for regional water resources management [25,48-50].”

  1. Here are some papers you can use them. Groundwater depletion in the Middle East from GRACE with implications for transboundary water management in the Tigris–Euphrates–western Iran region. Water Resour. Res. 2013, 49, 904–914. Integrated Geophysical Assessment of Groundwater Potential in Southwestern Saudi Arabia, Frontiers in Earth Sciences, 10.3389/feart.2022.937402.

A: Thanks for this suggestion. Both the two references are in GWS using GRACE, we add the references in Page 2, line 50.

  1. Please add some geological backgrounds to the Study area section.

A: Thank you for your advice. We added a description of the geological background of the study area and cited some literature. For detail, please check in Page3, line 102-109.

  1. Why did not use other sources of GRACE mascon data such as JPL, and GSCF.

A: Thanks for this question.

The GRACE mascon solutions are provided by three institutions, University of Texas Center for Space Research (CSR), Jet Pripulsion Laboratory (JPL) and Goddard Space Flight Center (GSFC). The CSR and JPL mascon solutions can be used directly without leakage corrections (Swenson and Wahr, 2006), Landerer and Swenson (2012). CSR solutions generally showed superior performance with mean RMS of 2.01 cm, less than that of JPL (3.19 cm) (Chen et al., 2021). Therefore, the mascon data from CSR were used in this study.

  1. Figure 1. Could you please adjust the vertical scale to let us see the variations in the time series?

A: Thank you for your advice. We have modified Figure 1 to ensure that its vertical coordinates are consistent. For detail, please check in Page 7, line 198.

Finally, thank you again for reviewing the manuscript.

Best wishes from all authors

Round 2

Reviewer 1 Report

The comments were considered and applied appropriately in the revised manuscript.

Reviewer 3 Report

Thank you for updating the paper.

Well done!